# Fluid-Induced Vibration of a Hydraulic Pipeline with Piezoelectric Active Constrained Layer-Damping Materials

**Yuanlin Zhang** [1], **Peixin Gao** [1,*] , **Xuefeng Liu** [1], **Tao Yu** [1,*] **and Zhaohua Huang** [2]

1   School of Electromechanical and Automotive Engineering, Yantai University, Yantai 264005, China; zhanglintf1207@outlook.com (Y.Z.); liuxuefeng1121@163.com (X.L.)
2   National Key Laboratory for Nuclear Materials, Nuclear Power Institute of China, Chengdu 610041, China; hzhnm@sohu.com
*   Correspondence: peigaox@ytu.edu.cn (P.G.); taoyuyt@126.com (T.Y.)

**Abstract:** The basic structure of a pipeline is complex due to the narrow installation space of a pipeline system. Thus, a considerable number of complex pipelines are adopted in a pipeline system. When a hydraulic pipeline works, it is impacted by fluid, which produces vibration. It is necessary to implement an effective method to control the vibration of a pipeline system. In recent years, the research on active constrained layer damping (ACLD) technology is increasing. However, there are few studies on the vibration characteristics of the ACLD pipeline system conveying fluid. The damping and vibration characteristics of ACLD pipeline system conveying fluid are studied in this paper. Considered the influence of the fluid–structure interaction, the motion equations can be derived, and the finite element model established of the pipeline based on ACLD treatment. The effect of the elasticity modulus, the thickness of the viscoelastic and constrained layer, the length and position of the ACLD patch, the velocity and pressure of fluid, and the voltage for the constrained layer, are all considered. The results show that ACLD technology has great damping influence on the conveying fluid pipeline.

**Keywords:** pipeline system; active piezoelectric constrained layer damping technology; fluid; finite element

## 1. Introduction

The hydraulic pipeline system, as an important energy transmission system, mainly provides fuel and lubricant; it is widely used in aerospace, ships, nuclear power, and other fields. Due to limited working space, a considerable number of L-shape pipelines are used in the hydraulic pipeline system. Pipeline vibration problems (caused by fluid–structure coupling and resonance) has attracted widespread attention from scholars. It is generally believed that the central causes of pipeline system vibration are due to i) the coupling effect of the components and substances in the hydraulic system, and ii) the complex coupling vibration of the hydraulic fluid.

Vibration reduction technology of a pipeline system could be summarized as passive and active methods. Passive constraint layer damping treatment (PCLD) does not demand external power and utilizes the pipeline motion to generate a control force [1]. Gao [2] and Zhai [3] reduced the vibration of a hydraulic pipeline by optimizing the layout of the pipeline clamps. A topology optimization method for stent layout optimization is proposed in the paper [4]. Its aim was to maximize the natural frequency of the structure. Xin [5] proposed an optimized layout method for the aircraft hydraulic piping system to reduce the vibration of the piping. Based on the finite element method, Gao [6] studied the damping and vibration characteristics of an aircraft pipeline system. In order to reduce the vibration of the pipeline, a viscoelastic material was added to the pipeline, and the influence of the additional mass and damping method on the vibration of the damping material was studied [7]. Ishikawa et al. [8] designed a disc-shaped viscoelastic

damping material connected to the pipeline to control the vibration of the pipeline. Metal rubber vibration isolators were designed to reduce the vibration of aero-engine pipelines in high temperatures and severe vibration environments [9,10]. These isolators have good vibration suppression effects for high-frequency vibration, but for low-frequency vibration, the vibration reduction effect is not obvious, and the control parameters cannot change with the change of the excitation environment. The main purpose of an active controller is to design different controllers that have strong adaptability, can sense the vibration changes of the pipeline in time, and provide good vibration suppression effects on low-frequency vibration. Compared with passive technology, the active constrained layer damping technology can actively adjust the shear deformation of the viscoelastic layer to suppress high-frequency vibration, as well as provide effective active control force to suppress low-frequency vibration (so it has wide frequency domain vibration control effects) [11].

The ACLD treatment was mainly used in beams and shells in the early stage. Guo [12] discussed the influences of parameters, such as the temperature, the angular velocity, and the gains of PD control on modal natural frequencies and damping ratios of the smart beam. Li [13] developed a comprehensive dynamic model of thin plate covered with ACLD patches, and he investigated the influence of control gain and acceleration on frequency and loss factor. Extensive efforts [14–16] have been exerted to optimal design of passive and active constrained layer damping treatment for vibrating structures. The performance of a segmented hybrid active-passive damping of the cantilever beam, compared with the passive beam, when some parametrics, such as the thickness of the viscoelastic layer and treatment length, were analyzed [17]. In [18], the authors discussed the effects of treatment length, constraining layer thickness and stiffness, and viscoelastic material thickness. There are few research studies on the application of ACLD technology in pipelines. In [19], the vibration control of the pipeline system using the active constrained layer damping treatment was investigated, in terms of the vibration and stress distribution. Zhang et al. [20] discussed the influence of structural parameters and the voltage applied to the constrained layer on the straight pipeline.

In [21], the problem surrounding active suppression of geometric nonlinear vibration of a fluid-conveying cantilever tube, under critical velocity, was reviewed. The finite element method was used to study the natural frequency of the fluid–structure interaction in a pipeline on a viscoelastic foundation [22]. Cravero [23] developed a CFD procedure to predict the tonal noise accurately. Various turbulence models have been tested to find the best results, without using too much computational resources. A three-dimensional simulation of the unsteady flow of the whole impeller-volute configuration was performed [24]. In [25], the researchers designed an effective method to predict the pressure fluctuation and dynamic response of pipeline fluid.

The finite element method is widely adopted in the dynamic modeling of pipeline. Lee et al. [26] used the finite element method to establish the dynamic equation of the fluid pipeline, and analyzed the effect of the fluid velocity on the stiffness matrix and damping matrix. Sreejith [27] established the dynamic model of the pipeline system by using the finite element method, and verified the effectiveness of the finite element method through the vibration response of the pipeline excited by a water hammer. The finite element model of the clamped pipeline system was established in [28], the authors verified the model effectiveness through testing. By using the Timoshenko beam theory, Chai [29] established a finite element model of the L-type pipeline with clamps, and verified the model by hammering and shaker tests. Gao et al. [30] established the finite element model for a pipeline system, developed an efficient long-distance solution procedure, and multi-support pipelines with elastic hoops, commonly used in aircrafts, in order to simplify the vibration analysis.

Although there is a vast amount of research on pipeline system vibration reduction, there is little research on the ACLD complex pipeline. This paper establishes the finite element model and derives the dynamic equations of the L-type pipeline system convey-

ing fluid, and studies the influence of structural and control parameters on the pipeline damping effect. This paper provides a technical and theoretical basis for active constrained layer damping control technology for complex pipeline systems conveying fluid.

## 2. The Finite Element Model of ACLD Pipeline

Figure 1 presents a finite element model for an L-type pipeline with a partially covered ACLD patch. The pipeline system has fixed boundary conditions at both ends. The pipeline is divided into two symmetrical parts (parts 1 and 2). *x1* is the distance from the ACLD patch to the left end of the pipe. *x2* denotes the length of ACLD patch. θ is the bending angle of pipeline. The finite element model is established based on the following assumptions: (a) constitutive materials of the pipeline are linear, isotropic, and homogeneous; (b) rotary inertia is negligible, and the shear deformations in the base beam and the piezoelectric layer are neglected; (c) the transverse displacement is the same for the three layers; (d) compared to those of the base pipeline and piezoelectric layer materials, the Young's modulus of the viscoelastic is negligible; (e) the interface is perfect continuity, and no slip occurs between the layers; (f) the viscoelastic layer only has the transverse shear, but no normal stress; (g) linear theories of elasticity, viscoelasticity, and piezoelectricity are used; (h) the fluid velocity profile is uniform and the fluid is inviscid and incompressible.

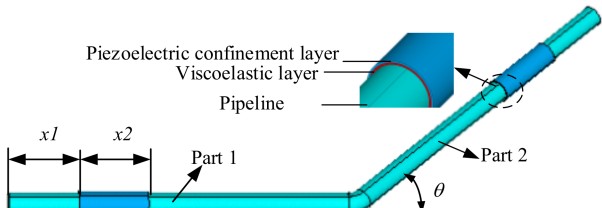

**Figure 1.** The finite element model for L-type pipeline.

### 2.1. Displacement Fields

The deformation of the pipeline with active constrained layer damping is presented in Figure 2. The pipeline and viscoelastic layer equations use Timoshenko beam elements according to the basic idea of the finite element method. The axial displacement of the neutral axis of the piezoelectric constraining layer and the base pipeline are $u_c$ and $u_b$, respectively. $h_b$, $r_b$, $h_s$, $r_s$, $h_p$, and $r_p$ are the thickness and the radius of the base pipeline, viscoelastic layer, and the piezoelectric constrained layer. $w$ denotes the transverse displacement. The shear strain and the shear angle of the viscoelastic layer are $β$ and $ψ$. $γ$ is the rotation of the viscoelastic layer from normal to the mid-plane.

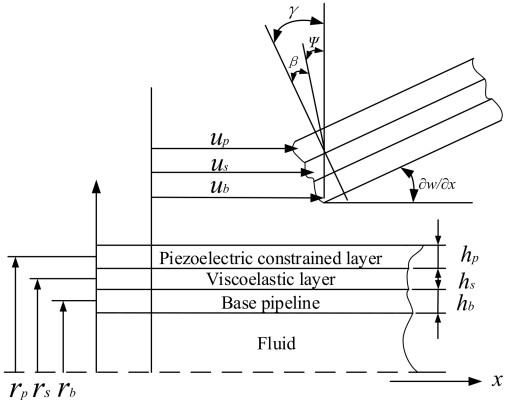

**Figure 2.** Deformation of the ACLD pipeline.

From Figure 2, shear strain $\beta$ in the viscoelastic material can be obtained by

$$\beta = \gamma - \psi \tag{1}$$

Based on above assumptions, in displacement and strain fields, one could see the constitutive relation of the pipeline system with an active constrained layer damping motion. Since the displacement distribution of viscoelastic layer is linear, the interlaminar displacement field are obtained by

$$u_p = u_s + \frac{h_s}{2}\left(\gamma - \frac{\partial w}{\partial x}\right) - \frac{h_p}{2}\frac{\partial w}{\partial x} \tag{2}$$

$$u_b = u_s - \frac{h_s}{2}\left(\gamma - \frac{\partial w}{\partial x}\right) + \frac{h_b}{2}\frac{\partial w}{\partial x} \tag{3}$$

The axial displacement and shear strain of viscoelastic layer are gained from the Equations (2) and (3)

$$u_s = \frac{1}{2}\left[(u_p + u_b) + \frac{h_p - h_b}{2}\frac{\partial w}{\partial x}\right] \tag{4}$$

$$\gamma = \frac{1}{h_s}\left[u_c - u_b + (h_s + \frac{h_b + h_c}{2})\frac{\partial w}{\partial x}\right] \tag{5}$$

*2.2. Subsection*

The FEM (Finite element method) is used to solve the dynamic equations of the pipeline. One ACLD pipeline element is described as a Timoshenko beam with two nodes (*i* and *j*), and is proposed in the present paper. Nodal displacements of the ACLD pipeline element are given by

$$\{Q^e\} = \left[w_i, w'_{i,}, u_{pi}, u_{bi}, w_j, w'_j, u_{pj}, u_{bj}\right]^{\mathrm{T}} \tag{6}$$

The axial displacement of piezoelectric constrained layer and the transverse displacement, the axial displacement of base pipeline are expressed in the nodal displacements by finite element shape functions

$$w(x) = [N_w(x)]\{Q^e\} \quad u_p(x) = [N_p(x)]\{Q^e\} \quad u_b(x) = [N_b(x)]\{Q^e\} \tag{7}$$

here $N_w(x)$ is the transverse shape function, $N_p(x)$ is the axis shape function of piezoelectric constrained layer, $N_b(x)$ denotes the axis shape function of baseline. And they are can be written as

$$N_w(x) = \left[1 - 3\left(\frac{x}{l_0}\right)^2 + 2\left(\frac{x}{l_0}\right)^3 \quad x - 2\frac{x^2}{l_0} + \frac{x^3}{l_0^2} \quad 0 \quad 0 \quad 3\left(\frac{x}{l_0}\right)^2 - 2\left(\frac{x}{l_0}\right)^3 \quad -\frac{x^2}{l_0} + \frac{x^3}{l_0^2} \quad 0 \quad 0\right] \tag{8}$$

$$N_p(x) = \left[0 \quad 0 \quad 1 - \frac{x}{l_0} \quad 0 \quad 0 \quad 0 \quad \frac{x}{l_0} \quad 0\right] \tag{9}$$

$$N_b(x) = \left[0 \quad 0 \quad 0 \quad 1 - \frac{x}{l_0} \quad 0 \quad 0 \quad 0 \quad \frac{x}{l_0}\right] \tag{10}$$

The shear strain and axial displacement of the viscoelastic layer can be represented as

$$\gamma = [N_{\gamma s}]Q^e \quad u_s = [N_{us}]Q^e \tag{11}$$

where

$$[\boldsymbol{N}_{\gamma s}] = \begin{bmatrix} 6h\left(-\frac{x}{l_0^2} + \frac{x^2}{l_0^3}\right) & h\left(1 - 4\left(\frac{x}{l_0}\right) + 3\left(\frac{x}{l_0}\right)^2\right) & 1 - \frac{x}{l_0} & -1 + \frac{x}{l_0} \\ 6h\left(\frac{x}{l_0^2} - \frac{x^2}{l_0^3}\right) & h\left(-2\left(\frac{x}{l_0}\right) + 3\left(\frac{x}{l_0}\right)^2\right) & \frac{x}{l_0} & -\frac{x}{l_0} \end{bmatrix} \tag{12}$$

$$[\boldsymbol{N}_{us}] = \begin{bmatrix} \frac{3(h_p - h_b)}{2}\left(-\frac{x}{l_0^2} + \frac{x^2}{l_0^3}\right) & \frac{h_p - h_b}{4}\left(1 - 4\left(\frac{x}{l_0}\right) + 3\left(\frac{x}{l_0}\right)^2\right) & \frac{1}{2}\left(1 - \frac{x}{l_0}\right) & \frac{1}{2}\left(1 - \frac{x}{l_0}\right) \\ \frac{3(h_p - h_b)}{2}\left(\frac{x}{l_0^2} - \frac{x^2}{l_0^3}\right) & \frac{h_p - h_b}{4}\left(-2\left(\frac{x}{l_0}\right) + 3\left(\frac{x}{l_0}\right)^2\right) & \frac{x}{2l_0} & \frac{x}{2l_0} \end{bmatrix} \tag{13}$$

Here $h = h_s + \frac{(h_b + h_p)}{2}$.

### 2.3. Energy Expressions

2.3.1. The Energy Expressions of Potential Energy

Due to bending and extension, the potential energy of the baseline can be expressed as

$$\frac{1}{2}E_b I_b \int_0^{l_0} \left(\frac{\partial^2 w}{\partial x^2}\right)^2 \mathrm{d}x = \frac{1}{2}E_b I_b \{\boldsymbol{Q}^e\}^T \int_0^{l_0} \boldsymbol{N}_w(x)_{xx}^T \boldsymbol{N}_w(x)_{xx} \mathrm{d}x \{\boldsymbol{Q}^e\} = \frac{1}{2}\{\boldsymbol{Q}^e\}^T \boldsymbol{K}_{bb}^e \{\boldsymbol{Q}^e\} \tag{14}$$

$$\frac{1}{2}E_b S_b \int_0^{l_0} \left(\frac{\partial u_b}{\partial x}\right)^2 \mathrm{d}x = \frac{1}{2}E_b S_b \{\boldsymbol{Q}^e\}^T \int_0^{l_0} \boldsymbol{N}_b(x)_x^T \boldsymbol{N}_b(x)_x \mathrm{d}x \{\boldsymbol{Q}^e\} = \frac{1}{2}\{\boldsymbol{Q}^e\}^T \boldsymbol{K}_{be}^e \{\boldsymbol{Q}^e\} \tag{15}$$

The element stiffness matrices for baseline can be given as

$$\boldsymbol{K}_{bb}^e = E_b I_b \int_0^{l_0} \boldsymbol{N}_w(x)_{xx}^T \boldsymbol{N}_w(x)_{xx} \mathrm{d}x \quad \boldsymbol{K}_{be}^e = E_b S_b \int_0^{l_0} \boldsymbol{N}_b(x)_x^T \boldsymbol{N}_b(x)_x \mathrm{d}x \tag{16}$$

Here $E_b$ $S_b$ $I_b$ denote the Young's modulus, cross-section area and the second moment of the base pipeline.

Due to the shear strain, the potential energy for the viscoelastic layer can be written as

$$\frac{1}{2}G_s S_s \int_0^{l_0} \lambda^2 \mathrm{d}x = \frac{1}{2}G_s S_s \{\boldsymbol{Q}^e\}^T \int_0^{l_0} \boldsymbol{N}_{\gamma s}(x)^T \boldsymbol{N}_{\gamma s}(x) \mathrm{d}x \{\boldsymbol{Q}^e\} = \frac{1}{2}\{\boldsymbol{Q}^e\}^T \boldsymbol{K}_{\gamma s}^e \{\boldsymbol{Q}^e\} \tag{17}$$

where

$$\boldsymbol{K}_{\gamma s}^e = G_s S_s \int_0^{l_0} \boldsymbol{N}_{\gamma s}(x)^T \boldsymbol{N}_{\gamma s}(x) \mathrm{d}x \tag{18}$$

Due to bending and extension, the potential energy of the piezoelectric constraining layer can be expressed as

$$\frac{1}{2}E_p I_p \int_0^{l_0} \left(\frac{\partial^2 w}{\partial x^2}\right)^2 \mathrm{d}x = \frac{1}{2}E_p I_p \{\boldsymbol{Q}^e\}^T \int_0^{l_0} \boldsymbol{N}_w(x)_{xx}^T \boldsymbol{N}_w(x)_{xx} \mathrm{d}x \{\boldsymbol{Q}^e\} = \frac{1}{2}\{\boldsymbol{Q}^e\}^T \boldsymbol{K}_{pb}^e \{\boldsymbol{Q}^e\} \tag{19}$$

$$\frac{1}{2}E_p S_p \int_0^{l_0} \left(\frac{\partial u_p}{\partial x}\right)^2 \mathrm{d}x = \frac{1}{2}E_p S_p \{\boldsymbol{Q}^e\}^T \int_0^{l_0} \boldsymbol{N}_p(x)_x^T \boldsymbol{N}_p(x)_x \mathrm{d}x \{\boldsymbol{Q}^e\} = \frac{1}{2}\{\boldsymbol{Q}^e\}^T \boldsymbol{K}_{pe}^e \{\boldsymbol{Q}^e\} \tag{20}$$

where

$$\boldsymbol{K}_{pb}^e = E_p I_p \int_0^{l_0} \boldsymbol{N}_w(x)_{xx}^T \boldsymbol{N}_w(x)_{xx} \mathrm{d}x \quad \boldsymbol{K}_{pe}^e = E_p S_p \int_0^{l_0} \boldsymbol{N}_p(x)_x^T \boldsymbol{N}_p(x)_x \mathrm{d}x \tag{21}$$

2.3.2. The Energy Expressions of Kinetic Energy

The kinetic energy of the base pipeline in the transverse and axial motion can be expressed as

$$\frac{1}{2}\rho_b S_b \int_0^{l_0} \left(\frac{\partial w}{\partial t}\right)^2 \mathrm{d}x = \frac{1}{2}\rho_b S_b \left\{\dot{\boldsymbol{Q}}^e\right\}^T \int_0^{l_0} \boldsymbol{N}_w(x)^T \boldsymbol{N}_w(x) \mathrm{d}x \left\{\dot{\boldsymbol{Q}}^e\right\} = \frac{1}{2}\left\{\dot{\boldsymbol{Q}}^e\right\}^T \boldsymbol{M}_{bt}^e \left\{\dot{\boldsymbol{Q}}^e\right\} \tag{22}$$

$$\frac{1}{2}\rho_b S_b \int_0^{l_0} \left(\frac{\partial u_b}{\partial t}\right)^2 \mathrm{d}x = \frac{1}{2}\rho_b S_b \left\{\dot{Q}^e\right\}^T \int_0^{l_0} N_b(x)^T N_b(x)\mathrm{d}x \left\{\dot{Q}^e\right\} = \frac{1}{2}\left\{\dot{Q}^e\right\}^T M_{ba}^e \left\{\dot{Q}^e\right\} \qquad (23)$$

Here

$$M_{bt}^e = \rho_b S_b \int_0^{l_0} N_w(x)^T N_w(x)\mathrm{d}x \quad M_{ba}^e = \rho_b S_b \int_0^{l_0} N_b(x)^T N_b(x)\mathrm{d}x \qquad (24)$$

The kinetic energy of the viscoelastic layer during lateral and axial motion can be written as

$$\frac{1}{2}\rho_s S_s \int_0^{l_0} \left(\frac{\partial w}{\partial t}\right)^2 \mathrm{d}x = \frac{1}{2}\rho_s S_s \left\{\dot{Q}^e\right\}^T \int_0^{l_0} N_w(x)^T N_w(x)\mathrm{d}x \left\{\dot{Q}^e\right\} = \frac{1}{2}\left\{\dot{Q}^e\right\}^T M_{st}^e \left\{\dot{Q}^e\right\} \qquad (25)$$

$$\frac{1}{2}\rho_s S_s \int_0^{l_0} \left(\frac{\partial u_s}{\partial t}\right)^2 \mathrm{d}x = \frac{1}{2}\rho_s S_s \left\{\dot{Q}^e\right\}^T \int_0^{l_0} N_{uv}(x)^T N_{uv}(x)\mathrm{d}x \left\{\dot{Q}^e\right\} = \frac{1}{2}\left\{\dot{Q}^e\right\}^T M_{sa}^e \left\{\dot{Q}^e\right\} \qquad (26)$$

Here

$$M_{st}^e = \rho_s S_s \int_0^{l_0} N_w(x)^T N_w(x)\mathrm{d}x \quad M_{sa}^e = \rho_s S_s \int_0^{l_0} N_{us}(x)^T N_{us}(x)\mathrm{d}x \qquad (27)$$

The kinetic energy of the piezoelectric constraining layer in transverse direction motion and in axial direction motion can be expressed as

$$\frac{1}{2}\rho_p S_p \int_0^{l_0} \left(\frac{\partial w}{\partial t}\right)^2 \mathrm{d}x = \frac{1}{2}\rho_p S_p \left\{\dot{Q}^e\right\}^T \int_0^{l_0} N_w(x)^T N_w(x)\mathrm{d}x \left\{\dot{Q}^e\right\} = \frac{1}{2}\left\{\dot{Q}^e\right\}^T M_{pt}^e \left\{\dot{Q}^e\right\} \qquad (28)$$

$$\frac{1}{2}\rho_p S_p \int_0^{l_0} \left(\frac{\partial u_p}{\partial t}\right)^2 \mathrm{d}x = \frac{1}{2}\rho_p S_p \left\{\dot{Q}^e\right\}^T \int_0^{l_0} N_p(x)^T N_p(x)\mathrm{d}x \left\{\dot{Q}^e\right\} = \frac{1}{2}\left\{\dot{Q}^e\right\}^T M_{pa}^e \left\{\dot{Q}^e\right\} \qquad (29)$$

Here

$$M_{pt}^e = \rho_p S_p \int_0^{l_0} N_w(x)^T N_w(x)\mathrm{d}x \quad M_{pa}^e = \rho_p S_p \int_0^{l_0} N_p(x)^T N_p(x)\mathrm{d}x \qquad (30)$$

In the above formula, $\rho$ is the density, $S$ denotes the cross-section area.

### 2.3.3. The Virtual of Kinetic Energy

One can see the constitutive equation of the piezoelectric material with the one-dimensional structure under uniaxial loading

$$\begin{bmatrix} S \\ D \end{bmatrix} = \begin{bmatrix} S_{11}^E & \mathrm{d}_{31} \\ \mathrm{d}_{31} & \varepsilon_{33}^\sigma \end{bmatrix} \begin{bmatrix} \sigma \\ E \end{bmatrix} \qquad (31)$$

where $S$ is the mechanical strain, $D$ is the electric displacement, $\sigma$ is the mechanical stress, and $E$ is the electric field intensity. The elastic compliance constant and the dielectric constant are $S_{11}^E$ and $\varepsilon_{33}^\sigma$, $\mathrm{d}_{31}$ denotes the piezoelectric constant. The stress-strain relation can be gained by the constitutive relation

$$\tau = E_p(S - \mathrm{d}_{31}E) \qquad (32)$$

Here

$$E_p = \frac{1}{S_{11}^E}; E = \frac{V_p(t)}{t_p} \qquad (33)$$

The virtual work done by the induced strain (force) is

$$\delta W_p = 2\int_0^{l_0} \int_0^{2\pi} E_p \mathrm{d}_{31} V_p(t)\delta\left(\frac{\partial u_p}{\partial x}\right)\mathrm{d}\theta\mathrm{d}x = [\delta Q(e)]^T \{f_p\} \qquad (34)$$

Here

$$\{f_p\} = \int_0^{2\pi} E_p d_{31} V_p(t) [N_p(x)]^T d\theta \tag{35}$$

where $V_p(t)$ is the voltage applied to the piezoelectric constraining layer.

*2.4. Fluid Element*

Considering the fluid–structure interaction, the fluid stiffness, mass, damping element matrices are

$$\boldsymbol{K}^{fe} = \left(T - pS_f(1 - 2\lambda) - m_f v^2\right) \int_0^{l_0} \boldsymbol{N}_w(x)_x^T \boldsymbol{N}_w(x)_x \mathrm{d}x \tag{36}$$

$$\boldsymbol{M}^{fe} = m_f \int_0^{l_0} \boldsymbol{N}_w(x)^T \boldsymbol{N}_w(x) \mathrm{d}x + m_f \int_0^{l_0} \boldsymbol{N}_b(x)^T \boldsymbol{N}_b(x) \mathrm{d}x \tag{37}$$

$$\boldsymbol{C}^{fe} = c \int_0^{l_0} \boldsymbol{N}_w(x)^T \boldsymbol{N}_w(x) \mathrm{d}x + m_f v \int_0^{l_0} \left[\boldsymbol{N}_w(x)^T \boldsymbol{N}_w(x)_x - \boldsymbol{N}_w(x)_x^T \boldsymbol{N}_w(x)_x\right] \mathrm{d}x \tag{38}$$

where $T$, $p$ and $v$ denote the longitudinal tension, fluid pressure and fluid velocity, respectively. $S_f$ denotes the cross-section area of the fluid, the mass density of fluid is $m_f$, $c$ is the damping because of the friction with the surrounding fluid.

*2.5. Load Vector*

$$\delta W_d = 2 \int_0^{l_0} f_\mathrm{d}(x,t) \delta w(x,t) dx = \{\delta \boldsymbol{Q}(e)\}^T \{f_d^e\} \tag{39}$$

$W_d$ is the virtual work done by external interference, $f_d^e$ is the element external disturbing force.

It is often more convenient to consider the impact of such forces at the global level.

*2.6. Dynamic Equation of the ACLD Pipeline*

The final stiffness and mass matrices for the ACLD element pipeline considering the fluid element can be represented as

$$\begin{cases} \boldsymbol{K}^e = \boldsymbol{K}_{be}^e + \boldsymbol{K}_{bb}^e + \boldsymbol{K}_{\gamma s}^e + \boldsymbol{K}_{pb}^e + \boldsymbol{K}_{pe}^e + \boldsymbol{K}^{fe} \\ \boldsymbol{M}^e = \boldsymbol{M}_{bt}^e + \boldsymbol{M}_{ba}^e + \boldsymbol{M}_{st}^e + \boldsymbol{M}_{sa}^e + \boldsymbol{M}_{pt}^e + \boldsymbol{M}_{pa}^e + \boldsymbol{M}^{fe} \end{cases} \tag{40}$$

where $e$ presents the pipeline element, *fe* denotes the fluid element, $\boldsymbol{K}_{be}^e$ and $\boldsymbol{K}_{bb}^e$ are element stiffness matrices for baseline, $\boldsymbol{M}_{bt}^e$ and $\boldsymbol{M}_{ba}^e$ denote the element mass matrices for baseline. The element stiffness matrices and mass matrices of viscoelastic layer are $\boldsymbol{K}_{\gamma s}^e, \boldsymbol{M}_{st}^e$ and $\boldsymbol{M}_{sa}^e$. The element mass matrices of constrained layer are $\boldsymbol{M}_{sa}^e$ and $\boldsymbol{M}_{pt}^e$, the element stiffness matrices of constrained layer are $\boldsymbol{K}_{pb}^e$ and $\boldsymbol{K}_{pe}^e$. $\boldsymbol{K}^{fe}$ and $\boldsymbol{M}^{fe}$ denote the element stiffness matrices and mass matrices of fluid, respectively.

For the pipeline, the transformation from local coordinate system to global coordinate system must be carried out before assembling element mass matrix and element stiffness matrix. The transformation relationship between the local coordinate system *O-xyz* and the global coordinate system *O-XYZ* can be represented as

$$\boldsymbol{Q}^e = \begin{bmatrix} \boldsymbol{Q}_1^e \\ \boldsymbol{Q}_2^e \\ \vdots \\ \boldsymbol{Q}_m^e \end{bmatrix} = \boldsymbol{Tq} = \begin{bmatrix} \boldsymbol{T}_1 & & & 0 \\ & \boldsymbol{T}_1 & & \\ & & \ddots & \\ 0 & & & \boldsymbol{T}_1 \end{bmatrix} \begin{bmatrix} \boldsymbol{q}_1 \\ \boldsymbol{q}_2 \\ \vdots \\ \boldsymbol{q}_m \end{bmatrix} \tag{41}$$

where $T$ denotes the coordinate transformation matrix and $T_1$ is the node transformation matrix.

$$T_1 = \begin{bmatrix} \cos(\lambda_i) & \sin(\lambda_i) & & & & & & \\ -\sin(\lambda_i) & \cos(\lambda_i) & & & & & & \\ & & 1 & & & & & \\ & & & 1 & & & & \\ & & & & \cos(\lambda_i) & \sin(\lambda_i) & & \\ & & & & -\sin(\lambda_i) & \cos(\lambda_i) & & \\ & & & & & & 1 & \\ & & & & & & & 1 \end{bmatrix} \tag{42}$$

here $\lambda_i$ is the included angle between the $X$-axis in global coordinate system and the $x$-axis in local coordinate system, as shown in Figure 3.

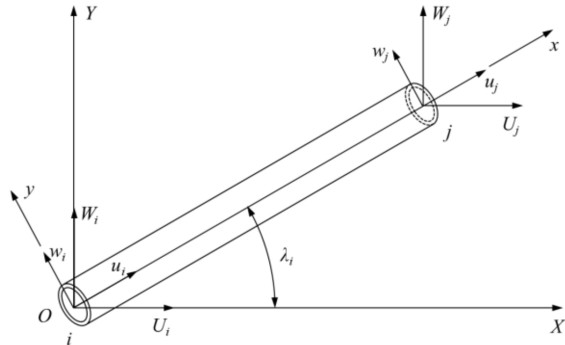

**Figure 3.** Relationship between local coordinate system and global coordinate system.

The stiffness matrix and mass matrix of the pipeline element in the global coordinate system is

$$\overline{K}^e = T^T K^e T, \overline{M}^e = T^T M^e T \tag{43}$$

The virtual work done by an external force can be expresented as

$$\delta W_{ex} = \int_0^{l_0} f_{ex}(x,t)\delta w(x,t)dx = \{\delta Q(e)\}^T \{f_{ex}\} \tag{44}$$

The stiffness, mass and damping element matrix are combined through the elements assembly, the dynamic equation of ACLD pipeline in the global system is obtained, it can be written as

$$M\ddot{Q} + C\dot{Q} + KQ = F \tag{45}$$

where $K$, $M$ and $C$ denote global stiffness, mass and damping matrices, respectively. $Q$ denotes the displacement vector, $F$ is the external force vector. For the undamped system, under free vibration, the dynamic equation of ACLD system is

$$\left[K - \Omega^2 M\right]\{Q\} = 0 \tag{46}$$

The natural frequencies $\omega$ and the loss factors $\eta$ of the ACLD pipeline can be represented as

$$\Omega = \omega^2(1 + i\eta) \tag{47}$$

$$\omega = (\mathbf{Re}(\Omega))^{\frac{1}{2}} \tag{48}$$

$$\eta = \frac{\mathbf{Im}(\Omega)}{\mathbf{Re}(\Omega)} \tag{49}$$

### 3. Validation

To verify the dynamic model of the pipeline established in the previous section, a numerical method and commercial software (ANSYS) were used to simulate the pipeline, respectively. For numerical analysis, the pipeline can be divided into 200 units on average. In the commercial method, the solid 45-element was used to model the base pipeline and viscoelastic layer. The piezoelectric material has a piezoelectric effect, and the piezoelectric effect analysis is a structure–electric field coupling analysis. A 3D solid element named SOLID5 was adopted to model the piezoelectric layer. SOLID5 has 3D magnetic, thermal, electrical, piezoelectric, and structural field capabilities, and the coupling between the fields are limited. The unit has eight nodes; each node has a maximum of six degrees of freedom. In the analysis of this paper, four degrees of freedom were used, namely the translational degrees of freedom UX, UY, UZ in the X, Y, and Z directions, and the voltage degrees of freedom VOLT. The geometrical and material properties of the ACLD pipeline are displayed in Table 1.

**Table 1.** Geometrical and material of ACLD pipeline.

| Quantities | Base Pipeline | Viscoelastic Layer | Constraining Layer |
|---|---|---|---|
| Elastic modulus (GPa) | 201 | – | 70 |
| Shear modulus (MPa) | – | 1 | – |
| Density (kg/m$^3$) | 7850 | 1580 | 2800 |
| Thickness (mm) | 2 | 0.5 | 0.5 |
| Poisson ratio | 0.3 | 0.498 | 0.3 |
| Loss factor | – | 0.29 | – |
| Pipeline outer diameter = 18 mm; the length of Part1 = 500 mm; the length of Part2 = 500 mm | | | |

Table 2 lists the comparison of the results of the natural frequencies and loss factors by the numerical method and ANSYS. A comparison of the first four shapes is shown in Figure 4.

**Table 2.** Natural frequencies (Hz) and loss factors of the first three modes by two methods.

| Mode | The Results by Numerical Method | | The Results by ANSYS | |
|---|---|---|---|---|
| | Modal Frequency (HZ) | Loss Factor | Modal Frequency (HZ) | Loss Factor |
| Mode 1 | 81.1697 | 0.0022 | 82.3188 | 0.0021 |
| Mode 2 | 222.8073 | 0.0038 | 222.5072 | 0.0036 |
| Mode 3 | 444.2120 | 0.0046 | 443.9939 | 0.0046 |
| Mode 4 | 896.3123 | 0.0109 | 897.1242 | 0.0108 |

From the data, it can be seen that the calculation results of the two methods are consistent. Therefore, the dynamics equation and finite element modeling of the pipeline in the previous section could be well verified. These results provide a theoretical and simulation basis for the parametric analysis of the pipeline.

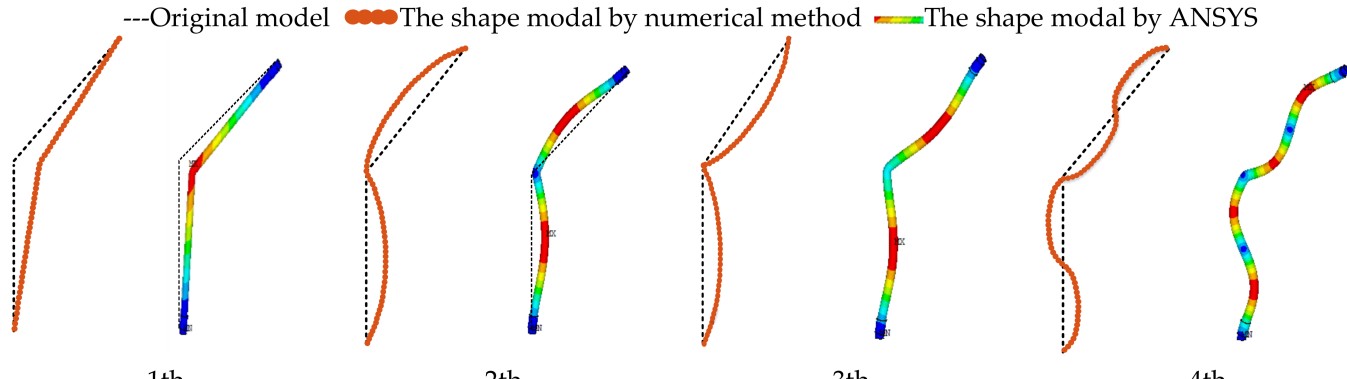

**Figure 4.** The comparison of the first four shapes in different ways.

## 4. Parametric Study and Discussion

### 4.1. Subsection

In the process of fluid flow in the pipeline, the fluid velocity changes due to the influence of pressure change, the pipeline, and other factors, resulting in vibration. Under the action of fluid velocity and pressure, the pipeline will produce complex lateral vibration, and the vibration of the pipeline will change the state of fluid movement. They interact with each other. Many previous pipeline vibration analyses, to a certain extent, ignored the influence of fluid factors on pipeline vibration. Although the simplified results could meet the engineering needs, they are not accurate enough. In this paper, considering the fluid effect in the pipeline, the finite element method is used to analyze and calculate the natural frequencies and loss factors of pipeline system.

Figure 5 shows the influence of natural frequencies and loss factors of bend, under different fluid velocity. It can be seen in Figure 5 that, with the increase of flow velocity, the stiffness of the pipeline decreases, resulting in the decrease of natural frequency, especially the first natural frequency. When the flow velocity reaches 350 m/s, the value is close to 0. At this time, buckling instability occurs in the pipeline. In the beginning, the loss factors of the pipeline increases slowly with the increase of flow velocity, but when the flow velocity is in the range of [330, 350], the first order loss factor increases sharply, and has little effect on the second and third mode loss factors.

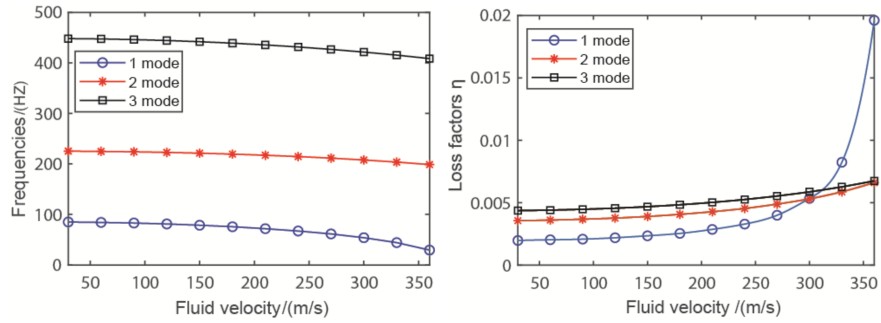

**Figure 5.** The influence of fluid velocity.

Figure 6 shows the variations of the natural characteristics of the pipeline under different fluid pressures. Similar to the effect of flow velocity, the natural frequency of the pipeline is reduced by fluid pressure, especially the first natural frequency, which tends to approach zero. The greater the fluid pressure is, the greater the loss factor is. The pressure has the greatest influence on the first-mode loss factor, and the first-mode loss factor increases sharply at [250, 300], in which the fluid pressure reaches the critical point.

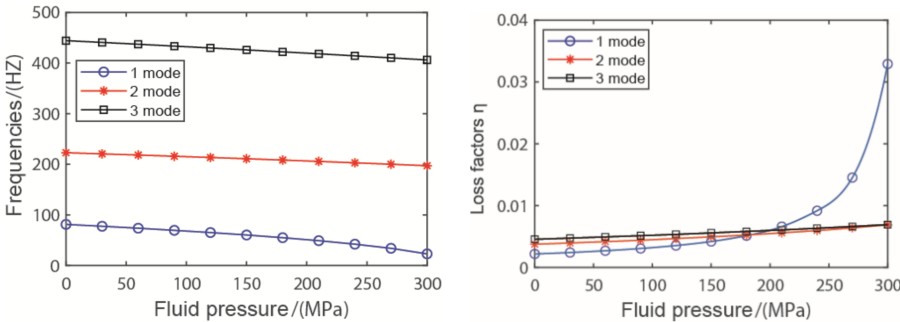

**Figure 6.** The influence of fluid pressure.

Overall, the fluid velocity and pressure have great influence on the natural frequencies and loss factors because of the fluid–structure interaction. In practical applications, in order to achieve the best damping effect, the velocity and pressure in the pipeline should be reasonably considered.

### 4.2. The Influence of the Viscoelastic Layer Parameters

The thickness of the viscoelastic layer is 0.25–3 mm; the effect of first three natural frequencies and loss factors of the pipeline are displayed in Figure 7. From the figure, the viscoelastic layer thickness has little influence on the frequencies compared with loss factors. The third-mode loss factor increases significantly from the beginning compared to first-mode and second-mode loss factors. The results show that the viscoelastic layer thickness can effectively improve the loss factors and vibration reduction effect of the pipeline.

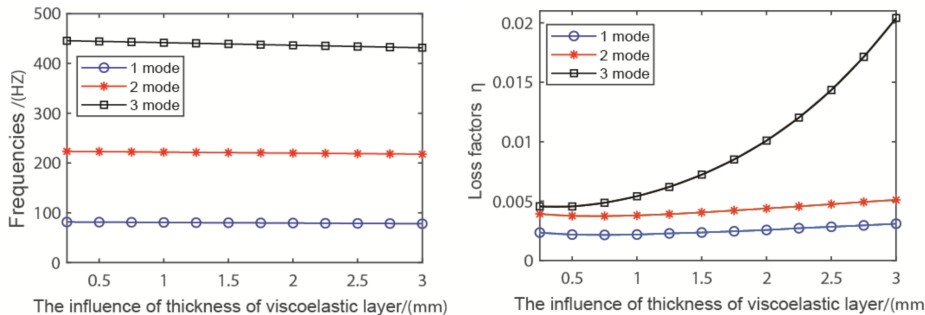

**Figure 7.** The influence of the thickness of the viscoelastic layer.

Figure 8 shows the impact of the elasticity modulus of the viscoelastic layer on the first three natural frequencies and loss factors of the pipeline. As the thickness of the viscoelastic layer has the same effect on the natural frequencies, shear modulus has a weaker effect on the natural frequencies, but has a greater impact on the loss factors because the stiffness of the pipeline changes little. The loss factor increases first and then decreases, and reaches the maximum at $10^7$–$10^8$. The maximum damping performance can be gained through the optimal value of the elastic modulus of the viscoelastic layer.

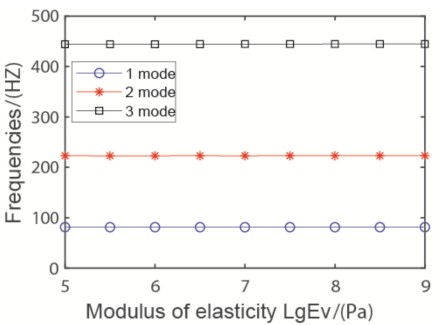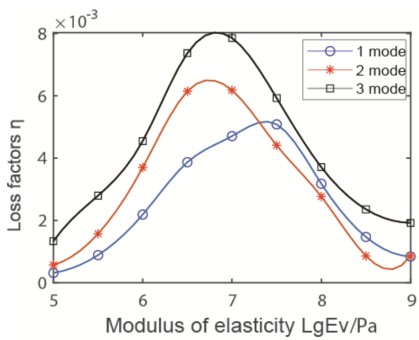

**Figure 8.** The effect of the modulus of elasticity of the viscoelastic layer.

### 4.3. The Influence of the Piezoelectric Confinement Parameters

The effect of the thickness of the piezoelectric confinement layer on the natural frequencies and loss factors of the pipeline is shown in Figure 9. The thickness of the constraining layer parameter varied from 0 to 4 mm. It can be seen from the figure that the thickness of the confinement layer has little effect on the natural frequencies of the pipeline. With the increase of the thickness of the confinement layer, the growth rate of the loss factor decreases gradually. The loss factor reaches its peak at the thickness of 2 mm, but its growth rate reaches its maximum at the thickness of 1 mm. In consideration of the economy, when the thickness of the confinement layer is 1–2 mm, it not only ensure the vibration reduction effect of the pipeline—it also save resources.

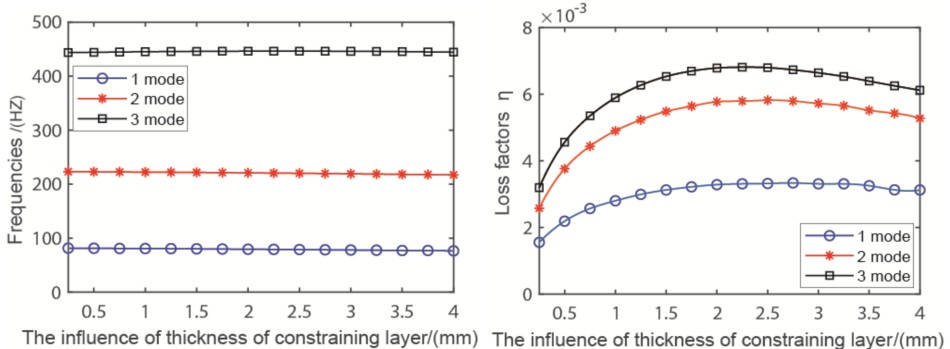

**Figure 9.** The influence of the thickness of the constraining layer.

The impact of the elasticity modulus of the constraint layer on the natural frequencies and loss factors of the system can be seen in Figure 10. Figure 10 shows that the increasing constrained layer elasticity modulus can significantly improve the damping performance at first. With the increase of the elasticity modulus of the constraint layer, the rising rate of loss factors gradually slows down. If the comprehensive damping effect and economy are considered, the young's modulus of the constrained layer is the best between 40 and 60 GPa.

### 4.4. Influence of the Length of ACLD Patch (x2)

As shown in Figure 11—take 0.1 m from the left end of the pipeline as the starting point, change the length of the ACLD patch, and analyze the natural frequencies and loss factors of the pipeline. There is almost no change in the natural frequencies, and the loss factors change greatly. With the increase of the length of the ACLD patch, the loss factor increases. Growth rate of the third-order loss factor is the largest compared to the first-order and second-order loss factors. The results show that the higher the coverage of the patch in the entire pipeline system, the better the damping effect.

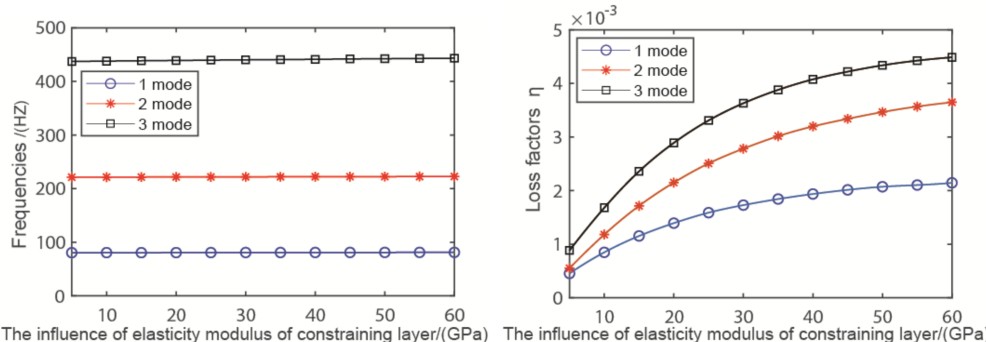

**Figure 10.** The effect of the elasticity modulus of the constrained layer.

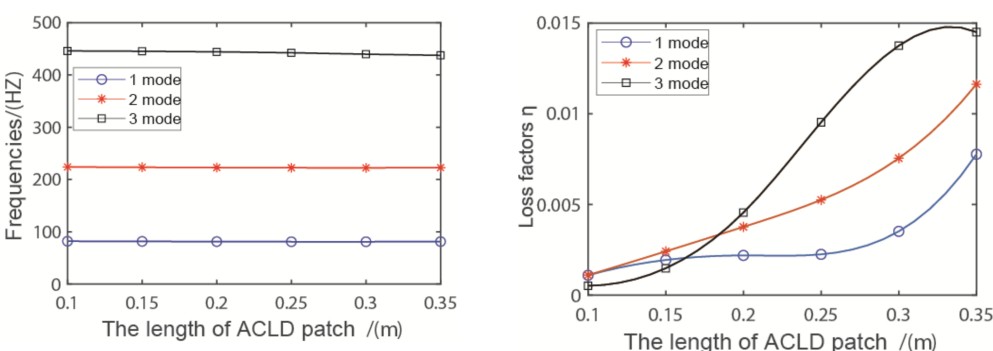

**Figure 11.** Influence of the length of the ACLD patch.

### 4.5. Influence of Position of the ACLD Patch (x1)

When the length of the ACLD patch (*x1*) remains constant for 0.2 m, the natural frequencies and loss factors of the pipeline are analyzed when the ACLD patch is in different positions. *x1* is shifted from 50 to 250 mm, from the extreme left end of the *Part1* pipeline (*Part2* is always symmetrical to *Part1*). From Figure 12, one can see the loss factors of the pipeline decreases first and then increases, and when *x1* = 150 mm, the loss factor has a minimum value, which indicates that the closer the ACLD patch is to the middle of the pipeline, the smaller the loss factor of the pipeline. When the ACLD patch is in the symmetrical position in Part1, the loss factor of the pipeline is very similar. In practical application, the position of the patch should be near both ends of the pipeline instead of the middle.

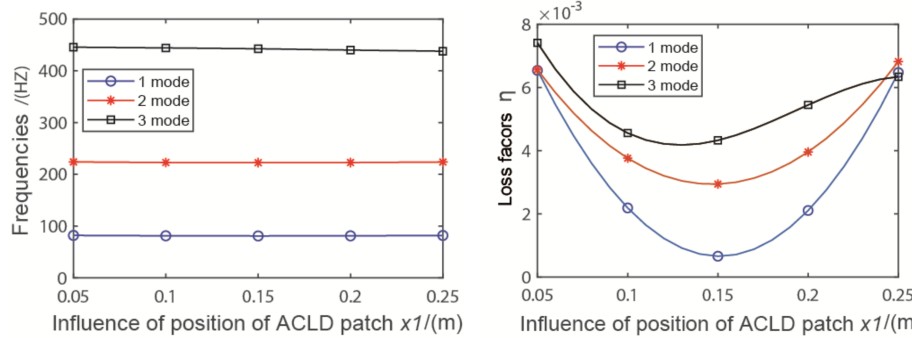

**Figure 12.** The influence of different position of ACLD patch.

### 4.6. Influence of the Angle (θ) of Pipeline

Influence of different angles on the natural frequencies and loss factors are presented in Figure 13. As the bending angle of the pipeline increases, the loss factors decline at high

speed, especially the second and third modes. It is clearly that the smaller the bending angle of the pipeline, the better the damping effect. Reducing the angle of the pipeline is helpful to reduce the vibration of the pipeline.

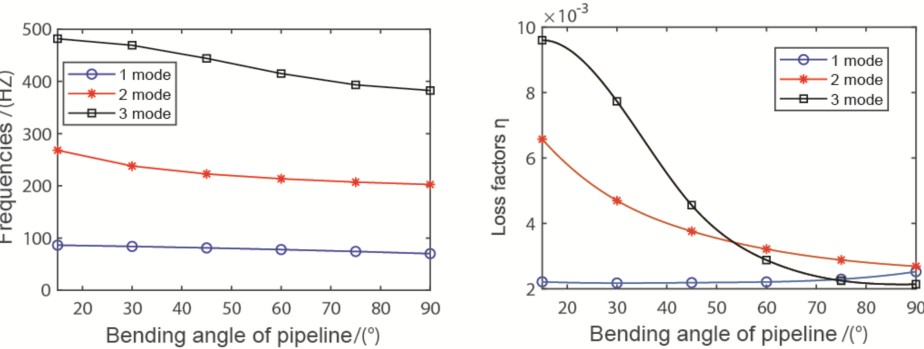

**Figure 13.** The influence of angle of pipeline.

### 4.7. The Influence of the Voltage

The effect of the ACLD patch for different voltages applied on the constrained layer with 0 V (PCLD), 30 V, and 50 V is presented in Figure 14. From the figure, it clearly shows that the loss factors of the active confinement layer-damping pipeline increase significantly, because, with the increase of the applied voltage, the shear deformation of the viscoelastic layer increases, which causes the vibration energy dissipation to become stronger.

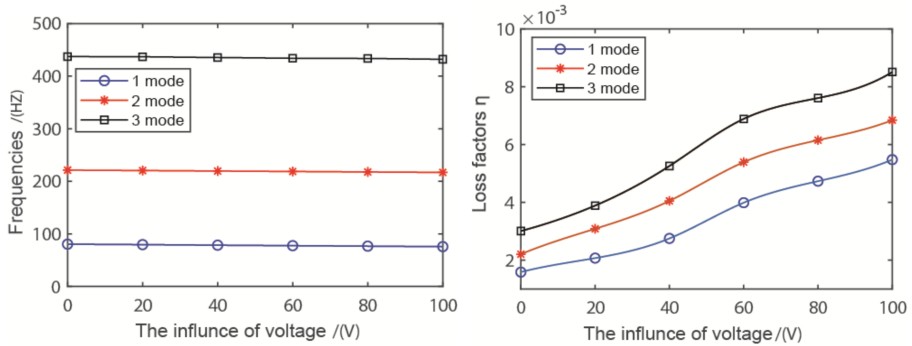

**Figure 14.** The influence of the voltage for the piezoelectric confinement layer.

## 5. Conclusions

The effectiveness of the ACLD treatment to reduce vibration of the pipeline system was verified in this paper. The motion equation of the ACLD pipeline system conveying fluid is derived by the finite element method. A "good agreement" of the result was obtained by the numerical method and commercial method. This paper discussed the influence of fluid-, viscoelastic layer-, and piezoelectric layer-related parameters to the vibration of the pipeline. Conclusions based on the current study are summarized as follows:

Due to the effect of fluid–structure coupling, the influence of fluid velocity and pressure on pipeline vibration is obvious. Increasing fluid velocity and pressure parameters could improve the damping characteristics of the pipeline, while reducing the natural frequency. It should be noted that, in the variation process of fluid velocity and pressure, there will be a critical point. In practical application, the occurrence of a critical state should be avoided to avoid buckling instability.

The best damping capability can be obtained by choosing suitable thickness of the constraining layer and viscoelastic layer. In addition, both the viscoelastic layer and piezoelectric constraining layer have the best elastic modulus values, which can make the vibration reduction effect reach the best state. By increasing the length of the ACLD patch,

the damping effect of the system will significantly improve. The smaller the bending angle of the pipeline, the better the vibration damping effect of the pipeline. With the increase of voltage applied for the constrained layer, the damping effect improves significantly.

Based on the above analysis, it is evident that the selection of parameters has an important impact on the vibration reduction effect of the pipeline. The results could be used when designing active vibration control applications for complex pipeline systems conveying fluid.

**Author Contributions:** Conceptualization, Y.Z.; methodology, Y.Z.; software, Y.Z.; validation, Y.Z. and X.L.; formal analysis, X.L.; investigation, X.L.; resources, P.G. and T.Y.; data curation, P.G.; writing—original draft preparation, Y.Z.; writing—review and editing, Y.Z. and X.L.; visualization, T.Y. and Z.H.; supervision, T.Y.; project administration, P.G. and Z.H.; funding acquisition, P.G. All authors have read and agreed to the published version of the manuscript.

**Funding:** This research was funded by the National Natural Science Foundation of China, grant number 51805462; national, major projects of aero-engines and gas turbines (J2019-I-0008-0008), and the Key Research and Important Technologies Development Program of Shandong Province, grant number 2019JZZY020114.

**Institutional Review Board Statement:** Not applicable.

**Informed Consent Statement:** Not applicable.

**Data Availability Statement:** The data presented in this study are available upon request from the corresponding author.

**Conflicts of Interest:** The authors declare no conflict of interest.

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
