# Peer review of "Fluid-Induced Vibration of a Hydraulic Pipeline with Piezoelectric Active Constrained Layer-Damping Materials"

_coatings, doi:10.3390/coatings11070757_

Round 1
Reviewer 1 Report
The paper’s interest is given by the combination of Kirchhoff-Love beam-shells in layered media with visco and piezo materials, with fluid in beam. It is relatively original but requires some light improvements detailed in the attached document.

Reviewer 2 Report
In this paper the effectiveness of the ACLD treatment to reduce vibration of pipeline system has been verified. The topic is quite interesting and in the complex the paper is well organized. The introduction can be improved by highlighting the contribute of the first 10 references that are only grouped. Then I suggest to insert some references about the fluid dynamic behaviour on similar topics, for example through the CFD is possible to analyse the aeroacoustics derived by flow interaction, as in the followings papers:
- Velarde S., Tajadura R., Numerical simulation of the aerodynamic tonal noise generation in a backward-curved blades centrifugal fan, Journal of sound and vibration, vol. 295, pp. 781-786, 2006.
- Cravero, C.; Marsano, D. “Numerical prediction of tonal noise in centrifugal blowers”. Turbo Expo 2018: Turbomachinery Technical Conference & Exposition, June 11-15, 2018, Oslo, Norway, ASME Paper GT2018-75243.
These references can give a more general character to the paper.
The numerical section is very complete and the validation section gives credibility to the model. However can you highlight in a table the boundary conditions used?
The result section is good with each parameter influence analysis well organized in each subsection.
The conclusions must be improved, to highlight the main results obtained.
